# Effects of Mixed of a Ketogenic Diet in Overweight and Obese Women with Polycystic Ovary Syndrome

**DOI:** 10.3390/ijerph182312490

**Published:** 2021-11-27

**Authors:** Raffaele Ivan Cincione, Francesca Losavio, Fabiana Ciolli, Anna Valenzano, Giuseppe Cibelli, Giovanni Messina, Rita Polito

**Affiliations:** 1Department of Clinical and Experimental Medicine, University of Foggia, 71100 Foggia, Italy; ivan.cincione@unifg.it (R.I.C.); losaviofrancesca@yahoo.it (F.L.); fabycioll@gmail.com (F.C.); anna.valenzano@unifg.it (A.V.); Giuseppe.cibelli@unifg.it (G.C.); rita.polito@unifg.it (R.P.); 2Department of Advanced Medical and Surgical Sciences, Università degli Studi della Campania “Luigi Vanvitelli”, 80138 Napoli, Italy

**Keywords:** polycystic ovary syndrome (PCOS), women, ketogenic diet (KD), adipose tissue (AT), obesity, insulin resistance, hormone production

## Abstract

Polycystic ovary syndrome (PCOS) is a commonly occurring endocrine disorder characterized by hirsutism, anovulation, and polycystic ovaries. Often comorbid with insulin resistance, dyslipidemia, and obesity, it also carries significant risk for the development of cardiovascular and metabolic sequelae, including diabetes and metabolic syndrome. The relationship between central obesity and the development of insulin resistance is widely verified. Adipose tissue excess and the coexistent dysregulation of adipocyte functions directly contribute to the pathogenesis of the metabolic complications observed in women with PCOS. In the light of these evidence, the most therapeutic option prescribed to obese women with PCOS, regardless of the phenotype e from the severity of clinical expression, is lifestyle correction by diet and physical activity. The aim of this study is to evaluate the beneficial effects of ketogenic diet in 17 obese women with PCOS. Our results showed that the ketogenic diet inducing therapeutic ketosis, improves the anthropometric and many biochemical parameters such as LH, FSH, SHBG, insulin sensitivity and HOMA index. In addition, it induces a reduction in androgenic production, whereas the contextual reduction of fat mass reduced the acyclic production of estrogens deriving from the aromatization in the adipose tissue of the androgenic excess, with an improvement of the LH/FSH ratio. This is the first study on the effects of the ketogenic diet on PCOS, however, further studies are needed to elucidate the mechanism underlying ketogenic diet effects.

## 1. Introduction

Polycystic ovary syndrome (PCOS) is the most common disorder of ovarian function and the most frequent cause of hyperandrogenism and anovulation in adolescents and adult women of fertile age, with a general prevalence ranging between 6 and 15% [1]. PCOS is characterized by hyperandrogenism, acne, hirsutism, disorders of menstrual, ovarian morphological alteration and by an increase in circulating levels of androgens, such as testosterone [2,3]. Genetic factors together with environmental, metabolic, and endocrine factors contribute to the multifactorial heterogeneity of the pathogenesis of this condition. Environmental factors are related to obesity and lifestyle habits, both playing an important role in PCOS [4]. Approximately 75% of PCOS patients are overweight and/or obese and central obesity is observed in both normal and in overweight women with PCOS [5,6,7]. Central obesity contributes significantly to the development of metabolic syndrome, with a prevalence in women with PCOS reported to be 43%, characterized by the presence of insulin resistance, hyperinsulinemia, and dyslipidemia [8]. The relationship between central obesity and the development of insulin resistance has been widely verified. Adipose tissue excess and the coexistent dysregulation of adipocyte functions directly contribute to the pathogenesis of the metabolic complications observed in women with PCOS. It is consolidated knowledge that adipose tissue is an endocrine organ, constituted not only by adipocytes but also by fibroblasts, macrophages, stromal cells, monocytes and stem cells such as pre-adipocytes [6], which are able to perform paracrine and endocrine functions through the production and secretion of molecules such as adipocytokines but also TNF-alfa and interleukin 6 (IL-6), the last is mainly secreted by activated macrophages resident in adipose tissue [6,7,8] and which are mainly increased in both subcutaneous and visceral fat. At the same time, the expansion of adipose tissue, as occurs in obesity, is caused by hypertrophy of the adipocytes and the consequent condition of hypoxia, which result in increased oxidative stress and autophagy [9]. Moreover, according to some reports, visceral obesity is very dangerous because it is able to alter adipose tissue endocrine functions, leading to an inflammatory cytokines production and inducing systemic low-grade inflammation know as [10,11]. Therefore, obesity, and not PCOS per se, is the main determinant of presence of this inflammatory status, but at the same time low-grade inflammation can be a cause of ovarian dysfunction with the arrest of folliculogenesis and consequent anovulation present in women with PCOS, also of insulin resistance itself. Hyperandrogenism and insulin resistance are two crucial aspects in the genesis of PCOS [5]. Abdominal adiposity and obesity could contribute to ovarian hyperandrogenism and adrenal also through mechanisms independent of insulin resistance. Indeed, adipose tissue acts on the local metabolism of sex steroids and cortisol in visceral fat [12] with an increased production of estrogen and estrone (with an increase of the estradiol/estrone ratio). At the hypothalamic level, the excess of estrone and androgens increases the pulsatility of GnRH secretion, with the preferential production of luteinizing hormone (LH) and hyperandrogenism.

In addition, the insulin resistance typical of obese women with PCOS, induces an increased risk of hyperandrogenism compared to normal weight patients [13]. The most recent unifying theory on the pathogenesis of PCOS focused on the pivotal pathophysiological role of insulin resistance and hyperinsulinemia. Insulin resistance, found in many patients with PCOS even at a young age, is independent of weight, but seems to have a relationship with total and central fat mass [14]. It is estimated that 50–70% of women with PCOS show a variable intensity of insulin resistance. The degree of hyperinsulinemia appears to correlate directly with the gravity of the syndrome and of obesity [15]. In light of this evidence, the most therapeutic option prescribed to obese women with PCOS, regardless of the phenotype and the severity of clinical expression, is lifestyle correction through diet and physical activity. Moreover, physical activity combined with a controlled diet positively regulate the maturation of follicles and improve more regular ovulation and pregnancy rates. [8,9,13,14]. Carbohydrate intake, especially with a high glycemic index and glycemic load, stimulates a rapid increase of blood glucose levels and, consequently, a compensatory increase in insulin production rate. Therefore, lowering the content of carbohydrates in food intake might have positive effects on PCOS more than standard hypocaloric diets, through the selective action on fat oxidation [12,13,14,15]. In this scenario, the ketogenic diet represents an effective approach in the treatment of metabolic imbalances due to insulin resistance and is therefore a possible treatment of PCOS in overweight or obese women. The ketogenic diet is a dietary protocol characterized by a reduced carbohydrate intake to about 40–50 g per day, a high intake of fats and a moderate amount of protein. The reduction of dietary intake of carbohydrates causes an increasing depletion of liver and muscle glycogen stores and a simultaneous reduction in glycemia and therefore also in the concentration of insulin [16,17]. This glycogen depletion is replaced by an increased oxidation of fatty acids derived from adipose tissue, through a reduced level of insulin which, in turn, leads to an increase in glucagon-induced lipolysis and the generation of ketone bodies, acetoacetate, β- hydroxybutyrate and acetone, which act as metabolic fuel [18,19]. Thus, ketosis reduces the need to produce endogenous glucose and can satisfy metabolic needs, sparing the use of amino acids from lean muscle tissue and using as main source of energy the lipids, inducing a rapid weight loss. For these reasons, the aim of this study is to evaluate the beneficial effects of ketogenic diet in obese women with PCOS.

## 2. Materials and Methods

### 2.1. Subject Selection

In this case, 17 overweight and obese women were enrolled at the University Medical Service of Dietetic and Metabolic Diseases of the Faculty of Medicine and Surgery of the University of Foggia. All patients provided written informed consent before the beginning of the study and after having had a detailed explanation of the tests. The study was approved by the local Ethics Committee on 22 May 2018, n°440/DS, and conducted according to the ethical principles of the Declaration of Helsinki. Written informed consent was obtained from all participants. The inclusion criteria were BMI > 25, age between 18 and 45 years (fertile age), diagnosis of PCOS according to the Rotterdam Criteria (oligo/amenorrhea or amenorrhea, clinical signs of hyperandrogenism and polycystic ovary), use of a previous diet consisting of at least 55% calories in the form of carbohydrates, willingness not to use contraceptives during the experimental period; desire to lose weight. The exclusion criteria were state of pregnancy and breastfeeding, kidney, liver or heart diseases, episodes of gout or hyperuricemia, suspension of estrogen-progestogen and/or insulin-sensitizer drug therapy for at least 12 months before the start of the dietetic protocol, and other etiologies (congenital adrenal hyperplasia, androgen-secreting tumors, Cushing syndrome) had been excluded.

### 2.2. Study Design and Settings

The dietary treatment protocol had a total duration of 45 days with the assessments of medical history, gynecological evaluation of oligo/amenorrhea status, nutritional status, body composition, and biochemical measurements at the first visit (baseline) indicated as time 0 (T0) and on the final visit at day 45 indicated as 1 (T1) as reported in Figure 1. The eligible subjects were assigned a diet, customized based on personal tastes and needs, but in compliance with the limit imposed by the need to keep the carbohydrate intake low. The prescribed foods were carefully chosen to ensure a balanced and varied diet, considering not only their energy value in quantitative terms but also their nutritional value in terms of quality. The protocol included the following assessments and outcome measures on the first visit (T0).

### 2.3. Anthropometric and Biochemical Parameters

Bodyweight was measured at each visit on the same scale with the subject wearing light clothing, but with shoes and socks removed, to the nearest 0.1 kg using an electronic scale with bioimpedance analysis (Tanita Model TBF-300A, Tanita Corp., Arlington Heights, IL, USA). Height was measured to the nearest 1 cm using a stadiometer (Seca 213, Seca), body mass index (BMI) was calculated in kg/m^2^, waist circumference was measured as the smallest circumference between the lowest rib and the iliac crest on the midaxillary line, hip circumference was measured at the level of the widest circumference over the great trochanters, waist to hip ratio was calculated as waist measurement divided by hip measurement [16]. Body fat mass (Fat Mass FM) expressed in kilograms and percentage, free fat mass (Free Fat Mass FFM) expressed in kilograms, total body water (TBW) expressed in kilograms, Muscle Mass (MM) expressed in kilograms, basal metabolism (BM) expressed in Kcal, (a rough estimate is provided) was analyzed using an electronic scale with bioimpedance analysis (Tanita Model TBF-300A, Tanita Corp.). All parameters were measured before and after the ketogenic diet protocol (T0 and T1).

Blood and urine samples were collected at the same session of visit. LH, follicle stimulating hormone (FSH), Insulin, C-peptide, Total Testosterone, and sex hormone binding globulins (SHBG) were measured by the immunochemiluminescent method, blood glucose by the enzymatic method with hexokinase, total cholesterol, HDL, LDLand triglycerides by the enzymatic colorimetric assay. Free Testosterone was measured by the radioimmunological assay. Homeostatic Model Assessment (HOMA) Index was calculated according to the formula (insulinemia µU/mL × glycemia mg/dL/405. A colorimetric method was used for serum albumin. Blood ketone 3-hydroxybutyrate (BHB), the most important indicator of ketosis, was assessed every day using Glucomen LX Beta Ketone (Menarini, Florence, Italy) on fresh capillary whole blood from the fingertip and on the same day urinary ketone acetoacetic acid was assessed by urinary stick (Ketostix Bayer). Gynecological evaluation of oligo/amenorrhea status was conducted. After the first evaluations (T0), the patients underwent a nutritional, body composition, and biochemical measurements, and a gynecological evaluation again at the end of treatment (T1), after 45 days from recruitment.

### 2.4. Dietary Treatment Protocol

The participants were assigned a diet, customized based on personal tastes and needs, but in compliance with the limit imposed by the need to keep carbohydrate intake low. The treatment protocol with total duration of 45 days, was formulated according to the criteria of ketogenic diet (KD), using a modified ketogenic diet protocol defined here as “mixed ketogenic”. The “mixed ketogenic” diet included a daily protein intake, in part, isolated whey protein powder derived from milk with a high biological value and complete amino acid composition profile with near zero carbohydrates and fat content, and partly from animal protein sources such as meat, fish, or eggs (Table 1). The formulation of the diet aimed to the establish ketosis, preserving the lean mass through the protein quota without being a high-protein diet. For this purpose, a protein intake of 1.1–1.2 g/kg/die ideal body weight was used. The maximum allowable intake of daily carbohydrates was set at 30 g, this was to reduce blood glucose quickly and significantly and therefore insulin levels along with the contextual increase in glucagon levels to obtain, through the consequent rapid activation of lipolysis, the hepatic production of ketones. The total daily caloric intake was established at around 600 kcal; thus, the diet was a very-low-calorie ketogenic diet, this induced a drastic reduction of calories aimed at obtaining weight loss mainly from the fat mass. The lipid component was set to 30 g/day, mainly consumed in the form of extra virgin olive oil in the amount of 10 g taken during the evening meal, to which the lipid quotas contained in meat and fish and in oil-dried nuts and oilseeds were added. Breakfast and lunch consisted of the above protein powder (isolated whey protein). A snack was based on oilseeds and oil-dried nuts such as almonds, walnuts, cashews, pistachios, and peanuts. The use of unsweetened drinks, such as herbal teas, tea, infused coffee, which were irrelevant for ketosis, were allowed. The protocol did not permit the use of food supplements and herbal extracts. The two-course dinner menu consisted of meat or fish, that the patient was able to choose daily, provided that fish was consumed at least three times a week. The portions of meat or fish were assigned based on individual protein requirements and body composition. The second course was based on vegetables. A list of vegetables was provided from which the patient was able to choose, and which included vegetables to be taken in free quantities because their limited carbohydrate component would not have exceeded the maximum daily carbohydrate intake of 30 g. The list of vegetables included all leafy vegetables: lettuce, valerian, Belgian salad, arugula, chicory, endive, escarole, beets, broccoli, thistles, cauliflower, cabbage, broccoli cabbage, cabbage, Chinese cabbage, cucumber, turnip greens, courgette flowers, fennel, mushrooms (champignons, ovules, porcini), bean sprouts, green peppers, radish, red and green radicchio, celery, spinach, and courgettes. Moreover, as mentioned above, 10 g of extra virgin olive oil was added to the dinner. Meat (or fish) was only replaced twice a week with eggs, in variable portions based on individual protein needs. Moreover, patients were asked to consume no less than 2 L of water/day. The exact bromatological composition of the diet was calibrated from subject to subject, where proteins represented on average 35–40% of daily calories, according to individual needs and body composition. Carbohydrates were, on average, 10–20 g per day, based on the number of vegetables consumed. The lipid content was on average 30 g of fat per day, most of which were of vegetable origin, taken through extra virgin olive oil, seed oil, and dried nut oil. It is essential to highlight that the ketogenic diet is not balanced from the point of micronutrients, and for this reason, multivitamin and multimineral supplements were administered throughout the mixed KD period to avoid nutritional deficiencies (Table 1).

### 2.5. Statistical Analyses

This was a pre-post, single-arm study. A two-tailed paired *t* test was used to test for statistical significance of the outcome variables at T0 and T1 using the software package StatView software 5.0.1.0. All data are expressed as mean ± standard deviation. The normality of data was assessed through the Pearson test. Significance was considered at a value of *p* < 0.05.

## 3. Results

### 3.1. Anthropometric Characteristics of Obese Women with PCOS at Baseline and after the KD Intervention

Our results showed a significant change in the anthropometric and biochemical parameters of KD obese subjects before and after diet therapy. First, anthropometric parameters such as weight and BMI were statistically reduced in KD obese subjects (Table 2). The mean age of the patients was 28.5 ± 5.38 years; the average weight at the first visit was 81.5 ± 13.56 kg, while the average BMI was 31.84 ± 5.85 kg/m^2^. Of the 17 patients, 12 (70.58%) reported the main symptom cycles lasting longer than 35 days, while only five patients (29.41%) had cycle absence. The patients were all diagnosed with PCOS, based on the ultrasound of typical ovarian morphology of the syndrome, with an increase in ovarian volume and/or an increase in small follicles. The clinical characteristics of the patients evaluated at baseline and after diet intervention are summarized in Table 2, Table 3 and Table 4.

The weight loss achieved with this protocol had an average difference between T0 and T1 of 9.4 kg (T0 91.85 ± 18.03 vs. T1 82.35 ± 17.17; *p* < 0.0001) corresponding to an average percentage loss of 9.4%, obtained with a nutritional intervention of only 45 days. Moreover, the mean BMI had a similar trend with an average drop between T0 and T1 of 3.6 kg/m^2^ (T0 34.75 ± 7.43–T1 31.14 ± 6.71; *p* < 0.001). The waist circumference was reduced, with an average difference value between T0 and T1 of 9.4 cm (T0 98.38 ± 10.45–T1 88.94 ± 10.72; *p* < 0.001) corresponding to an average percentage value of 9.6%, hip circumference was reduced with an average difference value between T0 and T1 of 8.1 cm (T0 118.9 ± 11.85–T1 110.8 ± 12.7; *p* < 0.001) and waist to hip ratio was reduced with an average difference between T0 and T1 of 0.02 (T0 0.82 ± 0.02–T1 0.80 ± 0.03; *p* < 0.001). Important results were obtained in fat mass reduction, FM expressed in kg, with an average loss between T0 and T1 of 7.90 kg (T0 40.32 ± 13.70–T1 32.41 ± 12.62; *p* < 0.001) corresponding to an overall loss of 4.58%. FFM expressed in Kg, showed a slight but statistically significant decrease from T0 to T1 of 1.41 (T0 52.86 ± 5.79–T1 51.45 ± 5.81; *p* < 0.05) also the value of MM expressed in Kg, showed a slight but statistically significant decrease from T0 to T1 of 1.32 (T0 50.18 ± 5.52–T1 48.46 ± 5.54; *p* < 0.05). TBW expressed in Kg appeared to be reduced from T0 to T1 by 1.32 (T0 38.42 ± 4.83–T1 37.10 ± 4.78; *p* < 0.01). If we consider TBW expressed as a percentage from T0 to T1 we have a negative value of −3.42 (T0 41.88 ± 3.89–T1 45.3 ± 4.70; *p* < 0.001) with a greater statistically significant value. Analyzing Basal Metabolic Rate, we found a statistically significant drop of 67 Kcal from T0 to T1 (T0 1694.33 ± 221.81–T1 1627.33± 217.09; *p* < 0.001) as shown in Table 2.

### 3.2. Metabolic Parameters of Obese Women with PCOS at Baseline and after KD Intervention

The mean levels of ketonemia and ketonuria increased over time with the continuation of the diet. All subjects at T0 had zero concentrations of blood and urine ketones while at T1 the mean blood concentration of ketones was 1.7 ± 0.58 mmol/L while the mean urinary concentration of ketones had an average value of 83 ± 54.34 mg/dL. The increase of ketones and ketonuria was statistically significant (*p* < 0.001).

The nutritional therapy strategy analyzed in this study also led to a net improvement in glycemic control: mean blood glucose (T0 95.21 ± 8.59–T1 85.14 ± 8.17; *p* < 0.001) had an average reduction of 10.07 mg/dL, blood insulin concentration of (T0 24.85 ± 22.18–T1 11.95 ± 7.59; *p* < 0.001) had a mean reduction of 12.90 µU/mL and c-peptide blood concentration had a significant decrease from T0 (2.88 ng/mL ± 1.36) to T1 (2.01 ng/mL ± 0.79;) with a mean reduction of 0.87 ng/mL (T0 2.88 ± 1.36–T1 2.01 ± 0.79; *p* < 0.001). The reduction in the Home Index (T0 6.05 ± 5.64–T1 2.60 ± 1.84; *p* < 0.05), with a mean decrease of 3.45. Serum Albumin increased (T0 3.97 ± 0.35–T1 4.28 ± 0.27; *p* < 0.001) with a mean value of–0.30 g/dL. Triglycerides were decreased with a mean value of 70 mg/dL (T0 270 mg/dL ± 44.29 T1 200 mg/dL ± 31.7; *p* < 0.001) as well as total Cholesterol that decreased (*p* < 0.001) by 40 mg/dL (T0 220 mg/dL ± 7.7 T1 180 mg/dL ± 5.8; *p* < 0.001) and LDL decreased by 35 mg/dL (T0 130.00 ± 21.1 T1 95.00 ± 5.3; *p* < 0.001) with an increase in HDL levels (T0 50 mg/dL ± 9.7–T1 65 ± 6.3; *p* < 0.001) (Table 3).

### 3.3. Endocrine Parameters of Obese Women with PCOS at Baseline and after KD Intervention

Plasma concentrations of LH had a significant variation (T0 11.56 mUI/mL ± 6.22–T1 6.58 mUI/mL ± 4.10; *p* < 0.001), with a mean decrease of 4.6 mIU/mL, free testosterone (T0 0.65 ng/dL ± 0.48–T1 0.47 ng/dL± 0.37) and total testosterone (T0 39.08 ng/dL±–T1 31.74 ng/dL ± 16.82) decreased by 0.17 ng/dLand 7.34 ng/dL, respectively, both were statistically significant (*p* < 0.001), and also the LH/FSH ratio decreased to 1.32 (T0 2.72 ± 1.30–T1 1.4 ng/dL ±0.96; *p* < 0.01). The FSH value increased in a significant manner (*p* < 0.05) with a mean value of 1.46 mIU/mL (T0 4.2 mUI/mL ± 1.76–T1 5.29 mUI/mL ± 2.40) increase. SHBG increase markedly (T0 54 nmol/L ± 40.27–T1 66.44 nmol/L ± 47.76) with a statistically significant (*p* < 0.001) mean value of 12.43 nmol/L as showed in Table 3. Endocrine parameters such as LH and FSH correlated positively with anthropometric parameters such as weight, BMI and FM, while LH/FSH negatively correlated with the same parameters (Figure 2). In addition, metabolic parameters such as glycemia positively correlated with LH and FSH, while LH/FSH negatively correlated with glycemia (Figure 3). Other parameters such as HOMA and insulinemia had the same trend as glycemia, however, the correlation was not strong (data not shown). Furthermore, SHBG negatively correlated with endocrinological parameters (Figure 3).

### 3.4. Gynecological Clinical Outcome of Obese Women with PCOS at Baseline and after the KD Intervention

The first and most important result of this protocol was the resolution of the irregularity of the menstrual cycle. Weight loss is, in fact, the first goal in the treatment of PCOS, not only for the crucial implications on the metabolic profile but also for the reproductive effects, numerous studies have shown how weight loss can ensure rapid recovery of the menstrual cycle. Of the 17 PCOS patients recruited, five patients had a natural reappearance of a regular menstrual cycle after years of amenorrhea and 12 patients benefited from menstrual cycle regularity restoration and five of the 12 patients achieved natural pregnancy after numerous previous unproductive attempts (Table 4).

## 4. Discussion

Polycystic ovary syndrome (PCOS) is the most common endocrine disorder in women of childbearing age. In PCOS an excess of androgens is often associated with metabolic disorders such as central obesity, insulin resistance, hyperinsulinemia, type 2 diabetes mellitus, and dyslipidemia. Insulin resistance has recently been proposed as the primary disorder from which other endocrine and reproductive anomalies of PCOS would arise, such as hyperandrogenism [20,21,22,23,24]. To date, there are no corrective treatments, although the use of some drugs can improve some clinical and biochemical parameters. The guidelines recommend a lifestyle change through diet and exercise as a first-line intervention in the management of PCOS [20]. The most accepted weight loss strategy is based on a simple reduction in daily calorie intake as part of a low fat/high carbohydrate diet, but there is still no clear data on which dietary protocols are most active, in the short and long term or even on what is the correct nutritional approach in general. Any therapy aimed at resolving or improving insulin resistance can undoubtedly guarantee an improvement not only from the metabolic point of view but also from the endocrinological point of view, being the two aspects most strongly interconnected [25]. In the last decade, interest in the ketogenic diet as nutritional treatment has increased as it is able to guarantee significant weight loss and to bring the countless benefits that derive from it, especially in insulin resistance conditions [26]. Our results showed that the KD has numerous beneficial effects in women affected by PCOS in a short period of 6 weeks (45 days). There was a strong amelioration of biochemical and anthropometric parameters such as weight loss, BMI, FM, glycemia, SHBG serum levels, and HOMA. Furthermore, these parameters also correlate with endocrinological hormones such as LH, FSH and LH/FSH.

The overproduction of LH with inversion of the LH/FSH ratio in favor of LH and the simultaneous decrease in the production of FSH mediated by the action of inhibin, is the basis of follicular maturation failure, anovulatory and the hyperstimulation of thecal cells by LH. As a consequence, we have an androgenic hypersecretion and the establishment of a self-fed vicious circle, to which insulin and IGF contribute, the latter locally produced in excess and enhanced by the often-coexisting condition of obesity, exerting a synergistic effect with LH in stimulating the synthesis of androgens with an increase in their circulation [27,28,29,30,31]. It should be remembered that hyperandrogenism is not in itself a cause of hyperinsulinemia, while the reverse is true. These hormonal changes are almost always associated with insulin resistance and consequent hyperinsulinism, of an important degree in about 1/3 of the patients, influenced by environmental factors, such as, first and foremost, obesity that is present in about half of the subjects with PCOS [1,2,3]. In fact, a reduced glucose tolerance or type 2 diabetes mellitus are the long-term sequelae of insulin resistance, especially in the presence of familiarity for type 2 diabetes mellitus, which has been observed in 45% of patients over 40 years old. The prevalence of metabolic syndrome is approximately double in women with PCOS than in the general population of the same age and BMI [12]. Given this pathophysiological characteristic, our dietary medical intervention, based on the administration of a mixed ketogenic diet during PCOS and considering the results obtained, can be a dietetic approach that may lead to a reversal of these processes. The results attest that the mixed dietary ketogenic medical intervention, by reducing the amount of carbohydrates and inducing therapeutic ketosis, acts favorably and effectively in the management of PCOS, in agreement with the study reported by Paoli et al., 2020, but in contrast with us, the authors reported a normo-caloric diet with the addition of phytocomplexes acting also on catecholaminergic system. In our study is reported the positive outcome of very low-calorie ketogenic diet in a short period and based on caloric restriction and its therapeutic ketosis related [32]. Furthermore, with the mixed ketogenic intervention, a reduction in blood glucose levels, insulinemia and an improvement in insulin sensitivity were obtained, which consequently led to a reduction in androgen production, whereas the contextual reduction of fat mass reduced the acyclic production of estrogens deriving from the aromatization in the adipose tissue of the androgen excess, with an improvement of the LH/FSH ratio. This last ratio improves through the reduction of the excess of LH thanks to a relative increase in FSH. No less important is that the mixed ketogenic diet resulted in an increase in albumin and SHBG, both free testosterone binding proteins, with a consequent reduction in bioactive free testosterone, thus contributing to a further improvement of hyperandrogenism. These data are very interesting and showed that this pathology is a multifactorial disease influenced by lifestyle. Indeed, as reported by Giampaolino et al., 2021, PCOS is influenced by the gut microbiota and then by nutrition [33]. In addition, a recent study reported the role of probiotics as modifiers of the gut microbiota in women affected by PCOS and then their potential role as drugs in PCOS treatment. These data suggested the important role of nutrition on the gut microbiota, on PCOS establishment and on their mutual relationship [34]. Furthermore, as showed by many data in the literature, non-diabetic women with PCOS and with associated insulin resistance showed lower levels of SHBG. Numerous studies have examined the relationships between SHBG and T2DM polymorphisms, suggesting that SHBG may represent a candidate gene for PCOS [35,36,37,38,39,40]. Moreover, the most interesting data are represented by the clinical outcome deriving from mixed ketogenic medical diet therapy: in fact, on the total number of women enrolled five patients had a natural reappearance of the menstrual cycle after amenorrhea, 12 patients benefited from the regularization of the cycle and five of these 12 patients achieved pregnancy after numerous previous unproductive attempts [41,42,43].

Therefore, mixed ketogenic medical diet therapy for only 45 days, intervening electively and specifically on some of the key pathophysiological moments that determine PCOS, such as insulin resistance, hyperinsulinemia, hyperandrogenism, the increase in LH and the inversion of the LH/FSH ratio, represents an adjuvant metabolic therapy in the management of PCOS, completely free of the adverse effects of the currently used drug therapies, with the added value of reducing the risk of developing obesity, metabolic syndrome and diabetic disease. No less important is the significant increase in terms of pregnancies achieved and regularization of the menstrual cycle [44]. Many data in the literature support our results, as reported by Gupta et al., the metabolic and endocrine effects of a low carbohydrate KD are evidenced by improvements in body weight, free testosterone percentage, luteinizing hormone/follicle-stimulating hormone ratio, and fasting insulin levels [15,16,17,43]. This leads to a decrease in androgen secretion and an increase in sex-hormone binding globulin, improving insulin sensitivity, and thereby normalizing endocrine functions. Such a dietary intervention and lifestyle management has beneficial effects in the treatment of PCOS patients affected with obesity and type 2 diabetes [45,46,47]. It has also been shown to improve depressive symptoms, psychological disturbances, and health-related quality of life in these patients [44,48,49,50].

## 5. Conclusions

The ketogenic diet has a positive effect on PCOS outcome. The positive effects obtained in a very short period represent a point of strength of our study. In addition, ketogenic diet is able to induce therapeutic ketonemia basing on caloric restriction, using only food. In this dietary protocol only calorie restriction is used and not the aid of functional foods that can induce weight loss. The use of caloric restriction in a short period represents a strong point of this ketogenic diet protocol able of inducing improvements in the outcome of PCOS. This diet intervention can improve the anthropometric and metabolic profile of these subjects. The ketogenic diet and its beneficial effects may be used as an adjuvant to pharmacological therapy for PCOS, in addition, in the first stage of this disease, the ketogenic diet may be useful as a non-pharmacological therapy, however, further studies are needed to clarify the molecular mechanism of the beneficial effects of the ketogenic diet.

## Figures and Tables

**Figure 1 ijerph-18-12490-f001:**
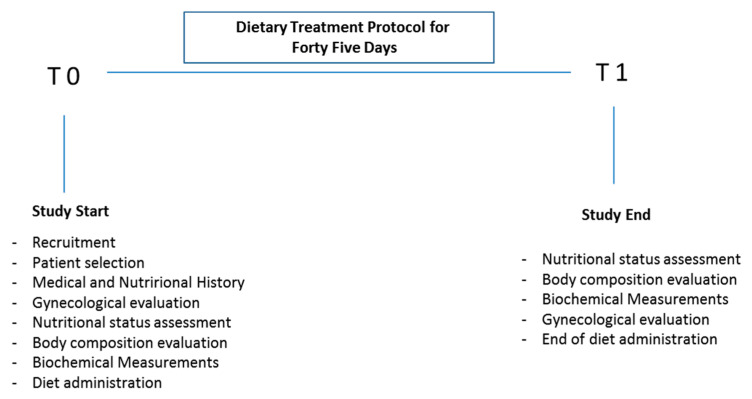
Experimental design: the obese women were recruited and underwent gynecological, anthropometric and biochemical assessments before and after the ketogenic diet protocol.

**Figure 2 ijerph-18-12490-f002:**
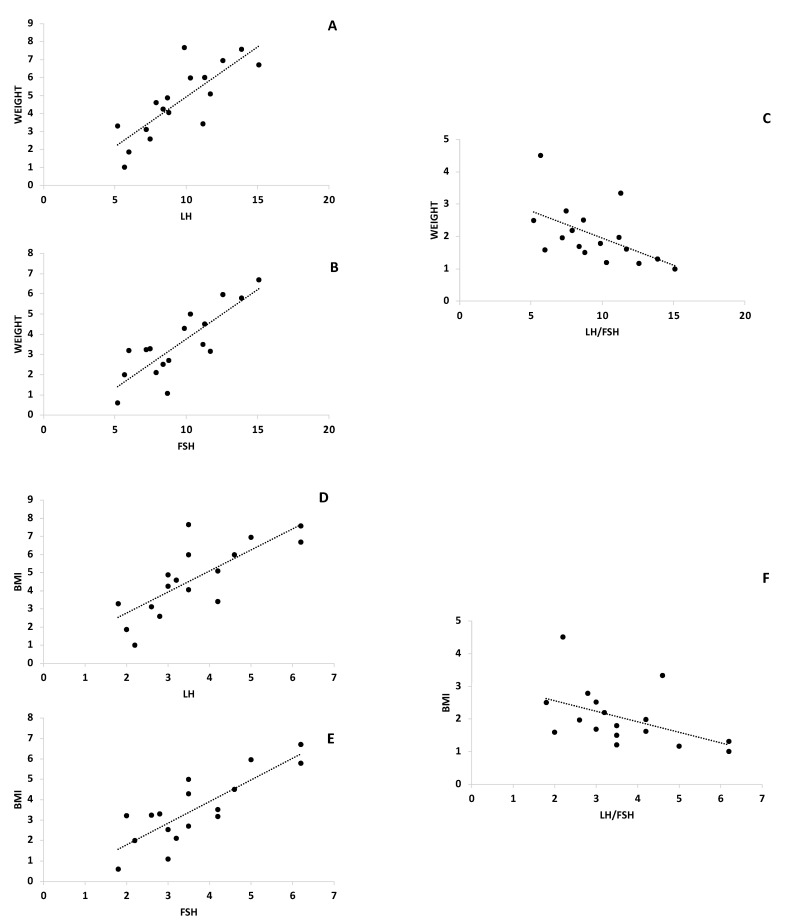
LH, FSH and LH/FSH strongly correlated with body weight, BMI and FM in KD subjects before and after diet. Δ-variation in KD subjects levels LH, FSH, LH/FSH serum levels and between body weight (panel **A**–**C**), BMI (panel **D**–**F**) and FM (panel **G**–**I**).

**Figure 3 ijerph-18-12490-f003:**
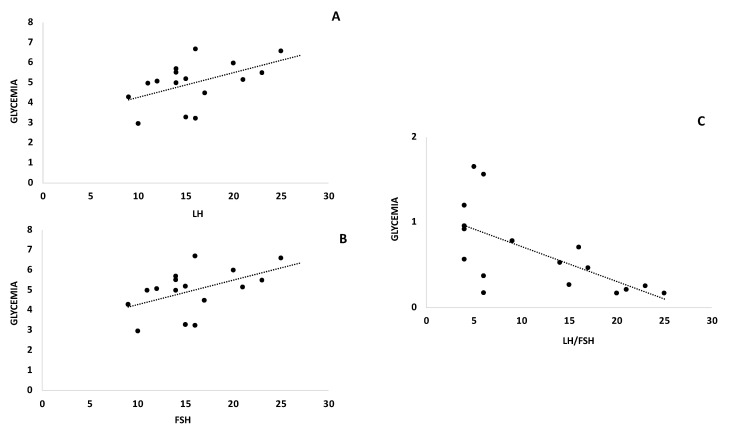
LH, FSH and LH/FSH correlated with glycemia and SHBG in the KD subjects before and after diet. Δ-variation in KD subjects levels LH, FSH, LH/FSH serum levels and between glycemia (panel **A**–**C**)) and SHBG (panel **D**–**F**).

**Table 1 ijerph-18-12490-t001:** Mixed Ketogenic Diet: carbohydrates, protein, fat and fiber composition.

Average Nutritional Values	100 g
Energy	1616 kJ/381 Kcal
Fat	1.69 g
Saturated fatty acids	1.48 g
Carbohydrates	1.40 g
Sugars	1.40 g
Fiber	
Protein	90.00 g
Salt	0.59 g

**Table 2 ijerph-18-12490-t002:** Anthropometric characteristics of PCOS obese women before and after the KD.

	T0	T1	
N = 17	MEAN ± SD	MEAN ± SD	*p*
Weight (kg)	81.52 ± 13.56	75.89 ± 13.31	<0.0001
BMI (Kg/m^2^)	31.84 ± 5.85	29.93 ± 5.47	<0.001
Waist circumference (cm)	98.38 ± 10.45	88.94 ± 10.72	<0.001
Hip circumference (cm)	118.94 ± 11.85	110.8 ± 12.7	<0.001
Waist to hip ratio	0.82 ± 0.02	0.80 ± 0.03	<0.001
FM (Kg)	40.32 ± 13.70	32.41 ± 12.62	<0.001
FFM (kg)	52.86 ± 5.79	51.45 ± 5.81	<0.05
Muscle Mass (kg)	50.18 ± 5.52	48.86 ± 5.54	<0.05
TBW (L)	38.42 ± 4.83	37.10 ± 4.78	<0.01
Basal Metabolic Rate (Kcal)	1649.33 ± 221.81	1627.33 ± 217.09	<0.001

**Table 3 ijerph-18-12490-t003:** Blood and urine parameters of PCOS obese women before and after KD.

	T0	T1	
N = 17	MEAN ± SD	MEAN ± SD	*p*
Triglycerides mg/dL	270.00 ± 44.29	200.00 ± 31.7	<0.001
Cholesterol mg/dL	220.00 ± 7.7	180.00 ± 5.8	<0.001
LDLmg/dL	130.00 ± 21.1	95.00 ± 5.3	<0.001
HDLmg/dL	50.00 ± 9.7	65.00 ± 6.3	<0.01
Glucose mg/dL	95.21 ± 8.59	85.14 ± 8.17	<0.001
Insulin μU/mL	24.85 ± 22.18	11.95 ± 7.59	<0.001
HOMA-IR	6.05 ± 5.64	2.60 ± 1.8	<0.001
Urinary ketones mg/dL	0	83 ± 54.34	<0.001
Blood ketones mmol/L	0	1.7 ± 0.58	<0.001
C-peptide ng/mL	2.8 ± 1.36	2.01 ± 0.79	<0.001
Serum Albumin g/dL	3.97 ± 0.35	4.28 ± 0.27	<0.001
LH mUI/mL	11.56 ± 6.22	6.58 ± 4.10	<0.001
FSH mUI/mL	4.29 ± 1.76	5.29 ± 2.40	<0.05
LH/FSH	2.72 ± 1.30	1.4 ± 0.96	<0.01
Free Testosterone ng/dL	0.65 ± 0.48	0.63 ± 0.66	<0.001
Total Testosterone	39.08 ± 20.88	31.74 ± 16.82	<0.001
SHBG nmol/L	54 ± 40.27	66.44 ± 47.76	<0.001

**Table 4 ijerph-18-12490-t004:** Gynecological clinical outcome of PCOS obese women before and after the KD.

N. of Patients	17		T0		T1
		Absence of cycle	5	Recurrent cycle	5
		Duration cycle > 35 days	12	Regularization of cycle	12
		Pregnancy	0	Pregnancy rate of Regularization of Cycle	5/12

## Data Availability

Data is contained within the article. Authors can use this data for research purposes only by citing our research article.

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
