# Peer review of "Effects of Mixed of a Ketogenic Diet in Overweight and Obese Women with Polycystic Ovary Syndrome"

_ijerph, 2021, doi:10.3390/ijerph182312490_

Round 1

Reviewer 1 Report

In this study, the authors show the effects of a ketogenic diet (KD) in overweight women with polycystic ovary syndrome (PCOS). Cincione et al. assessed anthropometric, metabolic and endocrine parameters as well as the gynaecological clinical outcome of PCOS women subjected to a ketogenic diet over a period of 45 days. As acknowledged by the authors, the effects of a KD on PCOS have been shown before, and using similar dietary protocols (references 38 and 39). Of note, the study of Paoli et al. 2020 “Effects of a ketogenic diet in overweight women with polycystic ovary syndrome.”  (https://doi.org/10.1186/s12967-020-02277-0) was not cited by the authors, but it is highly relevant.

The results in the Cincione et al. are convincing, but unfortunately they seem to only corroborate the literature. The results do not extend our current knowledge, functionally or mechanistically, on the benefits of a KD beyond anthropometric/metabolic/endocrine parameters, which are known to be modulated by a KD. The Cincione et al study did apply a specific dietary regimen: very low-calorie ketogenic diet (VLCKD). However, this is conceptually similar and, I would argue, of harder compliance when compared to other dietary KD protocols, such as a ‘eucaloric’ ketogenic regimen (Paoli et al 2020), which shows comparable benefits on BMI, body weight, and LH/FSH levels. Thus, the Cincione et al. study unfortunately does not show the novelty or insights expected of a research article, considering what is already known about KD in PCOS patients. I would encourage the authors to follow on these observations to further investigate the correlations/relationships found for some of the metabolic/endocrine parameters measured. The manuscript could also be improved in its writing and presentation.

Minor points:

The English can be improved; some sentences are confusing.

The hormones LH, FSH and SHBG are not spelled out, neither is the HOMA parameter.

Author Response

In attached the replay to review

Reviewer 2 Report

I read with great interest the enclosed Manuscript, which falls within the aim of IJERPH.

In my honest opinion, the topic is interesting enough to attract the readers’ attention. Methodology is accurate and conclusions are supported by the data analysis. Nevertheless, authors should clarify some point and improve the discussion citing relevant and novel key articles about the topic.

Authors should consider the following recommendations:

  • Manuscript should be further revised by a native English speaker since several non-English constructs and typos are notable
  • The Authors did not mention the sample size calculation for their study. It is essential to specify this data in order to guarantee an adequate significance of the results obtained by the Authors.
  • Please add the ID number of the ethical approval for this trial
  • The authors have not adequately highlighted the strengths and limitations of their study. I suggest better specifying these points
  • Does this manuscript conform the Enhancing the QUAlity and Transparency Of health Research (EQUATOR) network guidelines? It would be mandatory to declare about this element
  • In a multifactorial disease like PCOS, the role of microbiota has been reported as crucial by new shreds of evidence. Authors should discuss this key point by citing up-to-date literature (PMID: 33669557. PMID: 34194010)
  • The use of probiotics has been routinely promoted to increase the treatment's outcome. I suggest to discuss about this key-point (DOI: 10.2174/1573404816999200601162506)  
  •  
  •  

Author Response

in attached the replay to reviewer 

Round 2

Reviewer 1 Report

In reply to my comments, the authors state: “We thank the reviewer for these important suggestion, and we revised the text as suggested”. The authors extended the discussion and conclusion, having citied the Paoli et al. study, but have not addressed my main point: how is this study different from others studies showing the effect of a KD diet on PCOS? This study seems a corroboration of what has been published, so I suggest to discuss the strengths/benefits/differences of this study in more depth, for clarity.

The authors discuss:

“The ketogenic diet has a positive effect on PCOS outcome. The positive effects obtained in a very short period represent a point of strength of our study. This diet intervention can improve the anthropometric and metabolic profile of these subjects.” 

Was this mentioned because the Paoli et al. study applied a dietary treatment regimen of 12 weeks, while the Cincione et al. dietary treatment was for 45 days? If the ‘short treatment period’ is the main advantage of this study, can you clarify and discuss this more?

Secondly, is there any difference in the dietary protocol itself when compared to any other protocols cited? If there is a difference, did the authors have a rationale for it, in terms of the metabolic effects expected?

 The authors state a limitation (below), and I would be keen to also know the advantages.

Limitation of the study: this is one of the first studies investigating KD effects on PCOS outcome; however, a limitation to this study is the small number of patients considered. For this reason, it is our intention to expand the series in further studies on this subject.”

Alternatively, if what is important in this study is the corroboration/reproducibility aspect of KD diet effects in PCOS, please state and discuss it. Given the complexity and multifactorial nature of dietary effects, it is still important for different labs to reproduce dietary effects on a particular outcome.

Minor points:

Line 75: “knowed as [10-11].” Please, correct to ‘known as’, and add what is the thing that it is known as.

Lines 81-82: “Abdominal adiposity and obesity could contribute to ovarian hyperandrogenism e adrenal”. Please, correct ‘e’ to ‘and’.

Author Response

thank you 

Reviewer 2 Report

The authors have solved all the issues highlighted in the previous version of the manuscript.

I have no further concerns.

Author Response

thank you